# Evaluation of Switchgrass Genotypes for Cold-Tolerant Seed Germination from Native Populations in the Northeast USA

**DOI:** 10.3390/plants8100394

**Published:** 2019-10-02

**Authors:** Hilary Mayton, Masoume Amirkhani, Michael Loos, Jamie Crawford, Ryan Crawford, Julie Hansen, Donald Viands, Paul Salon, Alan Taylor

**Affiliations:** 1Horticulture Section, Cornell AgriTech, School of Integrated Plant Sciences, Cornell University, Geneva, NY 14456, USA; ma862@cornell.edu (M.A.); mtl72@cornell.edu (M.L.); agt1@cornell.edu (A.T.); 2Plant Breeding & Genetics Section, School of Integrated Plant Sciences, Cornell University, Ithaca, NY 14850, USA; jln15@cornell.edu (J.C.); rvc3@cornell.edu (R.C.); jlh17@cornell.edu (J.H.); drv3@cornell.edu (D.V.); 3United States Department of Agriculture/Natural Resources Conservation Service, Big Flats, NY 14830, USA; Paul.salon@ny.usda.gov

**Keywords:** switchgrass, C_4_ grasses, cold tolerance, bioenergy, seed germination, plant breeding

## Abstract

The focus of this research was to evaluate genotypes for cold-tolerant germination from wild switchgrass (*Panicum virgatum* L.) populations collected in the Northeast USA. Switchgrass nurseries were established in 2008 and 2009 with seed collected from native stands of switchgrass in the Northeast USA between 1991 and 2008. Switchgrass seed harvested from individual genotypes was evaluated for cold-tolerant germination in a series of laboratory experiments. Germination assays of seed of 13 switchgrass genotypes harvested in the fall of 2016 are the primary focus of this reported research. The selected genotypes were evaluated for cold-tolerant seed germination in three experiments, during the spring of 2017, fall of 2017 and spring of 2018, (with and without stratification) using a 10/15 °C regime with a 12 h photoperiod. Germination tests showed that several genotypes had significantly higher percentage germination as well as faster germination rates expressed as T50 (number of days required to reach 50% maximum germination) when compared to Cave-in-Rock, a moderately sensitive cold-tolerant commercial cultivar established in the original switchgrass nursery as a control. A final germination test was conducted to compare seed from the original population (no selection cycle 0), with one of the top performing cold-tolerant germination genotypes, and a commercial cultivar, ‘Espresso’, developed for low seed dormancy and low temperature germination. In this test, the selected genotype had significantly higher percentage germination in the stratified treatment and was not significantly different than Espresso in the non-stratified test. These data indicate successful selection for cold-tolerant germination in switchgrass genotypes from native germplasm collected in the Northeast USA.

## 1. Introduction

Switchgrass (*Panicum virgatum* L.) is a C_4_ open-pollinated and genetically diverse warm season perennial grass that can be used for forage, to improve the soil structure and health of overused and marginal farmlands, and as a bioenergy crop [1,2,3,4]. It is a self-incompatible species with nine chromosomes and an array of ploidy levels, however, most genotypes are either tetraploid or octaploid [5,6]. Switchgrass, big bluestem (*Andropogon gerardii* L.) and Indiangrass (*Sorghastrum nutans* L.) are the three major warm season perennial grasses that dominated the Great Plains of the mid-west USA and shaped the tall grass prairies [3,7]. Switchgrass is native to North America and wild populations can be found from Mexico to as far north as Montana and Southern Canada in the central North American continent and from Florida to Maine in the eastern regions of USA [8]. Phylogenetic analyses of native populations have shown that ploidy levels, overall genetic diversity, and physiological and morphological traits of switchgrass are related to adaptation to specific environments [5,9]. Generally, the upland northern ecotypes are shorter in stature and are more tolerant of cooler, drier environments than the lowland southern ecotypes, which are taller and grow well under high temperature regimes and wet soils [10]. In addition to adaptation response for vegetative and reproductive growth, studies on cardinal temperatures associated with seed germination also showed significant differences within upland and lowland genotypes [11]. The minimum, optimum, and maximum mean cardinal temperatures for seed germination for a set of diverse upland and lowland genotypes were 8.1, 26.6 and 45.1 °C, respectively. Seepaul et al. [11] developed thermotolerance classification categories for switchgrass genotypes. In their classification table, upland cultivars Shelter and Carthage were categorized as cold sensitive, Cave-in-Rock (CIR) was moderately sensitive, while Espresso was reported to be cold-tolerant [11]. 

Seed dormancy blocks the completion of germination and therefore may confound accurate assessment of germination, in particular testing for cold-temperature germination. Switchgrass dormancy is classified as non-deep, physiological dormancy is dormancy imposed by the embryo and seed covering layers [12]. The pericarp-testa is the primary morphological structure restricting switchgrass germination, while the lemma and palea also contribute to seed dormancy but to a lesser degree [13]. The morphology of these seed covering layers look similar for Espresso, therefore low dormancy of Espresso was attributed to embryo characteristics. Switchgrass seed dormancy is broken in the laboratory by moist chilling that is routinely conducted by placing seeds on moistened germination blotters at 5 °C for 14 days, termed stratification. Alternating temperatures also helps alleviate dormancy [14]. 

Switchgrass plants require three years to reach full maturity and to attain maximum yields, however once established, switchgrass stands can persist over twenty years, and yields can range from 2 to 25 t/ha depending on crop management and growing conditions [15,16]. A major impediment to production of switchgrass is the establishment of good first year plant stands [1,17,18], and soil moisture and temperature are critical factors that limit switchgrass germination and stand establishment. Conditions that specifically constrain good stand establishment in the Northeast USA are slow germination under cool wet soils, intense weed pressure and inherent high levels of seed dormancy [13,14,16,19]. Early seeding may provide a head start for switchgrass to out-compete weeds, however, cool spring soil temperatures often reduces seed germination [11]. Late seeding can provide additional time for more optimal soil temperatures and field preparation on marginal land, however, switchgrass also needs time for sufficient root growth to overwinter. Development of commercial cultivars that can germinate and thrive in cool soil temperatures common in early spring in the Northeast USA would help to broaden the planting window for switchgrass seeding in this region of the USA.

Maize (*Zea mays* L.) is one of the most widely grown C_4_ grasses in the world and is an excellent example of strategic breeding for rapid germination at low temperatures and production in regions with cool climates and short growing seasons. The availability of multiple genetically diverse populations allowed for successful screening and selection of maize genotypes with the ability to germinate under cold conditions [20]. Sorghum (*Sorghum bicolor* L.) is another C_4_ warm season grass with high genetic variation that has also undergone many rounds of selection and genetic screening for improved germination and growth in cool temperate climates [21]. 

Previous research on the importance of genetic diversity and local adaptation of switchgrass genotypes across environments have led to the development of plant breeding programs focused on improvement of cultivars for specific regions of the USA [1,2]. Due to the significant potential attributed to switchgrass as a dedicated energy crop by the USA Department of Energy (DOE) Biomass Program and an interest in the prospect for a new sustainable crop for growers in New York State (NY), a breeding program was initiated at Cornell University, Ithaca, NY in 2008. Seed for the switchgrass breeding program was collected by personnel from the United States Department of Agriculture (USDA) Plant Materials Center (PMC), Big Flats, NY, from native stands of switchgrass in the Northeast USA between 1991 and 2008 (Appendix A). Fifty native switchgrass accessions were planted in a nursery in 2008 and 31 additional native switchgrass accessions were planted in 2009 with the accessions collected from seven northeastern states (Appendix A). One goal of the breeding program and the focus of this study was to improve first year stand establishment though selection of genotypes for cold-tolerant germination from natural populations of switchgrass. Preliminary screening of seed for germination at different temperatures in the laboratory collected from individual plants in the switchgrass nurseries indicated that there was variation in the population for this trait. Therefore, the specific objective of this study was to evaluate several selected genotypes for cold-tolerant germination. The seed from cold-tolerant genotypes in this study were evaluated for germination with and without stratification using an alternating temperature to negate some influence of dormancy on germination. 

## 2. Results 

Germination assays of seed of 13 switchgrass genotypes harvested in 2016 are the focus of this reported research. The selected genotypes were evaluated for cold-tolerant seed germination in three experiments, during the spring of 2017, fall of 2017 and spring of 2018, (with and without stratification) using a 10/15 °C regime with a 12 h photoperiod. Switchgrass genotypes used in experiments are listed in Table 1. The Parent GRIN Number contains the precise location within the county where parent seed was collected. Additional information on Cycle 0 seed (seed obtained from the original population with no selection) can be found in Appendix A.

Data from the 2017 spring and fall germination tests (experiments 1 and 2) show that several genotypes had significantly higher overall maximum percentage germination (MPG) than the commercial cultivar CIR. In addition, many cold-tolerant lines had higher germination rates expressed as T50 (number of days required to reach 50% of maximum germination) than the CIR commercial check cultivar (not selected for cold tolerance) harvested from the switchgrass nursery (Table 2 and Table 3). In the first experiment, the four best cold-tolerant/low dormancy genotypes in the stratified treatment had 55, 44 and 40 MPG versus 3 percent for CIR, respectively (Table 2). In the non-stratified treatment, the data were similar as the two best genotypes had 20, and 18, MPG, while the CIR commercial check had 1% germination (Table 2). Germination rates (T50) were not significantly faster for the switchgrass lines than CIR in both stratified and non-stratified treatments (Table 2). Six of the nine switchgrass lines reached 50% germination after 10–12 days. The CIR check had low T50 values 12 and 11 days, respectively, for stratified and non-stratified treatments, however, the overall MPG was very low (3 and 1%, respectively) therefore the low T50 is inconsequential due to the fact that very few seeds germinated. In addition to increased germination, all but one of the cool temperature genotypes had significantly higher seed vigor index scores in the stratified assay in experiment 1 than CIR (Figure 1). The majority of switchgrass lines in the non-stratified assay also had higher seed vigor scores than CIR. Seed vigor is an important indicator for seedling survival and growth, which can contribute to better stand establishment.

Maximum germination results from the second controlled environment test conducted in the fall of 2017 were similar to the first experiment. The MPG ranged from 3–55% in the first experiment versus 4–67% in the second experiment for the stratified seed and 1–20% and 2–28% in non-stratified seed assays, respectively. All cold-tolerant switchgrass genotypes had significantly higher MPG than CIR after stratification (Table 3). In the non-stratified treatment, six out of 13 genotypes had significantly higher MPG than the commercial control check. The cold-tolerant line NY13 had the highest MPG in the both stratified (67%) and non-stratified (28%) assays and also had more rapid germination rate than the CIR check. The top-performing genotypes also had significantly higher seed vigor index scores than the commercial check, which is an indication of overall higher and more rapid seed germination (Figure 2).

Correlation analysis of MPG and T50 for genotypes evaluated for the stratified treatments in experiments 1 (spring 2017) and 2 (fall 2017) both showed positive and significant relationships with coefficient of variation (R^2^) values of = 0.75 and 0.78, respectively, indicating consistent germination across experiments after seed was stratified. Correlation analysis of non-stratified MPG and T50 for experiments 1 and 2 were not significant (R^2^ = 0.42 and 0.04 respectively). Lack of significant correlation for the non-stratified seed germination assays is not unexpected as switchgrass genotypes were not selected for low dormancy and, without stratification, switchgrass can have irregular germination due to inherent differences in dormancy. 

In the final experiment, seed from the original population (cycle 0) harvested in 2010 and 2011 was evaluated with one of the cold-tolerant genotypes (NY13), CIR, and Espresso (a line recently released for rapid germination at cool temperatures).

In the last experiment, the selected NY13 genotype had significantly higher MPG in both the stratified and non-stratified treatments than the non-selected CIR control check and the cycle 0 seed collected from the original switchgrass population (Table 4). NY13 also had significantly lower T50 and higher seed vigor index values than the cycle 0 seed and the control CIR (Figure 3). Maximum percentage germination for the genotype was significantly higher (89%) than Espresso (68%) in the stratified treatment but not significantly different than Espresso in the non-stratified treatment with 78% versus 83% MPG, respectively. Germination rates (T50) for the cold-tolerant genotype and Espresso were also not significantly different in both stratified and non-stratified treatments (Table 4).

All three cycle 0 seed controls had low MPG (<20%) and low seed vigor scores (Figure 3). Lastly, the switchgrass genotype NY13 and Espresso had the highest seed vigor index scores. Seed vigor index was similar for Espresso with and without stratification verifying that this variety had little dormancy, while genotypes from the cold-tolerant nursery were not specifically selected for low dormancy (Figure 3).

Maximum percentage germination for all genotypes in experiments 1 and 2 ranged from 3 to 67% in the stratified treatments and 1 to 28% in the non-stratified controlled environment tests. A tetrazolium (TZ) test conducted to determine if some genotypes had low viability revealed that genotype NY2, which had the lowest MPG in experiment 1, and NY1 with the lowest MPG in experiment 2 (Table 2 and Table 3), had 62% and 46% viable seed, respectively. Seed viability for the cultivar CIR, which had low MPG in experiments 1 and 2, showed a high level (84%) of viable seeds. 

The TZ test of the cycle 0 seed was inconclusive; however, a germination test conducted at standard AOSA temperatures of 20/30 °C [20] showed high levels of germination in both the stratified (92%, 95% and 98%) and non-stratified (64%, 66%, and 57%) experiments. The high percent germination for the cycle 0 seed at 20/30 °C temperatures with little germination under the cold temperature regime shows the successful progress made for cold-tolerant germination from the original switchgrass nursery.

## 3. Discussion

The focus of this research was on evaluation and selection of genotypes with potential to germinate at low temperatures from native switchgrass germplasm collected in the Northeast USA. Screening and evaluation of switchgrass germplasm for cold-tolerant germination began in 2010 and was completed in 2018. Five of the seven parent accessions used to establish a cold-tolerant germination nursery were originally collected from NY while one was collected in Pennsylvania and one from New Jersey. (Appendix A). In the final experiment reported in this study, the cold-tolerant germination genotype NY13 performed as well as the switchgrass cultivar Espresso, which was selected for cold-tolerant germination and low dormancy from a breeding program in Mississippi, USA [11].

A substantial amount of preliminary screening of the germplasm was conducted prior to the research discussed in this study. Research conducted on cold tolerance in maize, rice (*Oryza sativa)*, sorghum, barley (*Hordeum vulgare*), and many other agriculturally important crops have identified specific differentially expressed genes and quantitative trait loci involved with germination and growth at the seedling stage [22,23]. One study found that over expression of the maize dehydrin gene in tobacco resulted in more rapid germination at 15 °C [24]. Maize, rice and sorghum are much more economically important crops than switchgrass, however, due to its importance as a potential bioenergy crop and for use in land reclamation many genomic tools are now available for breeding improvement in switchgrass [1]. There are biparental and association mapping populations of both southern and northern germplasm, expressed sequence tags, marker-trait associations, a reference genome and linkage maps of both upland and lowland genotypes [25,26,27]. Breeding and selection for traits of interest in switchgrass are difficult due to the large genome, multiple ploidy levels and the perennial life cycle (three years to reach maturity). Use of genomic tools to identify specific genes or loci involved with cold-tolerant germination may reduce labor and time needed for evaluation of germplasm in the field and laboratory for development and release of new cultivars. No genetic screening and gene profiling was conducted in this study, therefore, the degree of genotypic variation in the switchgrass nurseries established in 2008 and 2009 and the cold-tolerant nursery established in 2014 are unknown. However, similar analyses on wild populations have shown high levels of genotypic and phenotypic diversity [27]. Evaluation of this population may improve the understanding of genetic mechanisms underlying seed germination at low temperatures in switchgrass. 

## 4. Materials and Methods

### 4.1. Switchgrass Genotypes

Switchgrass genotypes (individual plants) with cold tolerance were selected from breeding nurseries established in 2008 and 2009 in Ithaca, NY (Appendix A). Both nurseries consisted of sixteen plant plots of native accessions and commercial check cultivars planted in randomized complete block designs, replicated four times, with 0.9 m^2^ grid spacing between each plant. The soil type for the 2008 nursery is a Langford-Chippewa channery silt loam, while soil where the 2009 nursery was located is an Erie-Chippewa channery silt loam. Commercial check cultivars were Cave-in-Rock (CIR), Carthage, Kanlow, and Shelter (seed for check cultivars was purchased from Ernst Conservation Seed, Meadsville, PA, USA). 

Because of time and labor constraints, seed was harvested from every surviving genotype in the first replication of the 2009 switchgrass nursery in 2010, and from the tallest and most vigorous genotypes in both nurseries in 2011 (869 genotypes in total). In both years, the seed was cleaned but not stratified in the harvest year and 0.17 g (~100 seeds) from each genotype was placed on moistened brown blotter paper in petri dishes and transferred to a germinator with alternating temperatures (30/15 °C with 12 h photoperiod) to assess germination for 21 days. Two hundred and sixteen genotypes that were either tall and vigorous, or had greater than 45% seed germination in the first test were then tested in a germinator at a lower temperature of 20/15 °C with a 12-h photoperiod for 21 days. The relationship between a genotype’s seed germination at higher and lower temperatures is shown in Appendix A. Based on this preliminary testing, as well as height and vigor data for the genotypes, 47 genotypes were selected to establish a cold-tolerant population and moved to an isolated location for seed production. Seed from 44 of the 47 switchgrass genotypes (three had insufficient seed) were germinated in the spring of 2014 in the greenhouse and these half-sibling seedlings were used to establish a cold-tolerant space plant nursery in the field along with check control rows of commercial cultivars CIR, Carthage, and Shelter. 

The 2014 cold-tolerant space plant nursery consisted of eight plant plots from each of the 44 switchgrass half-siblings and commercial checks planted in a randomized complete block design with 0.9 m^2^ grid spacing between each plant. Each half-sibling plot was replicated 5 times, except for 9 half-siblings, as not enough plants were available. The soil type where the space plant nursery was located is an Erie-Chippewa channery silt loam. In 2015, 290 “best-plant-in-plot” selections were made based on plant vigor and height (data not shown). Seed was harvested from these selections and evaluated for cold tolerance in germination tests during the spring and summer of 2016. Seed was cleaned and prepared as described previously, and 0.17 g each of stratified and non-stratified seed from each genotype was tested at 20/15 °C with a 12-h photoperiod for 21 days. In these tests, genotypes with germination greater than 95% and with sufficient available seed for further testing and were selected for further analysis. In the spring of 2016, the selected genotypes were moved into an isolated seed production block and in the fall of 2016, seed from these genotypes was harvested and evaluated in this study. In the spring and fall of 2017, seed from 13 genotypes (Table 1) was evaluated under more stringent controlled conditions in three separate controlled environment assays for this study. All seed for experiments reported in this publication was preserved in a cold (5 °C) seed storage facility, located at Cornell AgriTech in Geneva, NY to maintain quality for the duration of the experiments from harvest in the fall of 2016 to the fall of 2018 except for cycle 0 seed harvested in 2010 and 2011, which was stored at 5 °C at a seed storage facility in Ithaca, NY.

### 4.2. Seed Germination Tests

Seed from switchgrass genotypes used in experiments reported in this study are listed in Table 1. The Parent GRIN Number contains the precise location within the county where parent seed was collected. Additional information on Cycle 0 seed (harvested from the original population with no selection) can be found in Appendix A. In addition to the genotypes, the cultivar CIR, which produced a sufficient amount of seed for germination tests, planted in the breeding nurseries in 2008 and classified as moderately sensitive for cold-tolerant germination, was used in all three germination experiments as a control to evaluate progress in selection [11]. The CIR commercial cultivar was not evaluated in the preliminary screening for cold tolerance and therefore serves as a non-selected control. During the spring and fall of 2017, seed from the selected genotypes, along with CIR seed collected from the nurseries, was evaluated for germination under cool temperatures with and without stratification. A final test was conducted which compared one of the best cold-tolerant genotypes with seed collected from the switchgrass nurseries in 2010 and 2011 (cycle 0) and a commercial cultivar ‘Espresso’, Plant Variety Protection number 201800200 (Personal communication, B. Baldwin). Dr. Brian Baldwin at Mississippi (MS) State University (Starkville, MS) developed Espresso for low dormancy and low temperature germination [11]. 

Seed for all experiments was cleaned using an air-column to remove low-density plant material and chaff. Four replications of clean seed of each switchgrass genotype were placed on moistened blue blotter papers in petri dishes with 50 seeds per plate. For the stratification treatment, all seeds were placed in a germinator for 14 days at 5 °C. After stratification, seeds were transferred to a germinator with alternating temperatures (10/15 °C with 12 h photoperiod) to assess germination. Non-stratified seeds were placed directly into the germinator under the same temperature and light cycles shown above. Germination counts were recorded weekly for each experiment from 7 to 42 days. Due to the difference in the age of the cycle 0 seed harvested in 2010 and 2011, a standard seed germination test [28] at 20/30 °C with a 16/8 h photoperiod, respectively, was conducted with the cycle 0 seed to evaluate seed quality. 

The maximum percentage germination was calculated using the number of germinated seeds/number of total seeds multiplied 100. Maguire developed the speed of germination [29] that combines both the percent and rate germination, and is referred to in this paper as the ‘Seed vigor index’. Number of days required to reach 50% maximum germination T50 was calculated using the following equation T50 = $ti+\left[\frac{\left(N+1\right)/2-ni}{nj-ni}\right]\;\left(tj-ti\right)$ [30]. Where N is the final number of germinated seeds, and ni and nj are total number of seeds germinated at time ti and *tj*, where ni < (*N* + 1)/2 < *nj*.

### 4.3. Tetrazolium Test

A tetrazolium test was used to evaluate the cycle 0 seed, the CIR check cultivar and a subset of cold-tolerant genotypes for viability. Seeds were sliced longitudinally and soaked in a 1% solution overnight (12 h) and were then placed in a germinator at 25 °C in the dark. The following day, seeds were evaluated visually and scored for viability as detected by TZ stain [31].

### 4.4. Data Analysis

All maximum germination percentage data from the three controlled chamber assays were evaluated for normality, homoscedasticity and goodness of fit using Proc Univariate-Normal (statistical software version 9.3, SAS Institute, Cary, NC, USA) along with generation of histograms of the distribution of residuals and normal probability plots. In addition, all maximum germination percentage data were subjected to arcsine square root transformation and evaluated for normality, homoscedasticity and goodness of fit using the same procedures described above. Non-transformed maximum germination percentage data had normal distributions in all three experiments and transformation did not improve normality or tests of goodness of fit. Therefore, non-transformed raw data were used to determine statistical differences for maximum germination percentage among plant genotypes. Non-transformed raw data for T50 (days to reach 50% germination) and seed vigor index data evaluated for normality and goodness of fit using the same procedures described above showed normal distributions and were homoscedastic and were therefore also not transformed in subsequent analyses. Analysis of variance (PROC GLM) was used to evaluate maximum germination percentage, T50 and seed vigor index data from experiments 1, 2, and 3 using SAS statistical software version 9.3 (SAS Institute, Cary, NC, USA). Determination of statistical significance of differences between means, was accomplished with the Fisher’s LSD test at a significance level of α=0.05. Correlation analysis (using raw data) was conducted on maximum percentage germination data and germination rates (T50) from experiments 1 and 2 to assess reproducibility and consistency of results using the statistical program JMP Pro 11 (SAS Institute, Cary, NC, USA).

## Figures and Tables

**Figure 1 plants-08-00394-f001:**
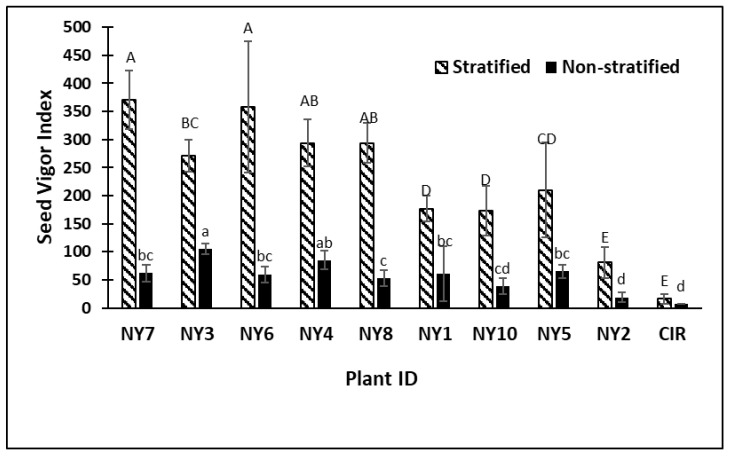
Experiment 1, spring 2017 results showing seed vigor index of switchgrass genotypes and the CIR check in stratified and non-stratified germination assays. Capitalized letters represent significant differences among Plant ID genotypes after seed stratification, while lower case letters represent significant differences for non-stratified Plant ID genotypes. Plant IDs with the same letter are not significantly different. Errors bars are standard deviation of means of four replications per Plant ID. *P* < 0.0001 value for both stratified and non-stratified germination with LSD values of 83.3 and 31.9 respectively.

**Figure 2 plants-08-00394-f002:**
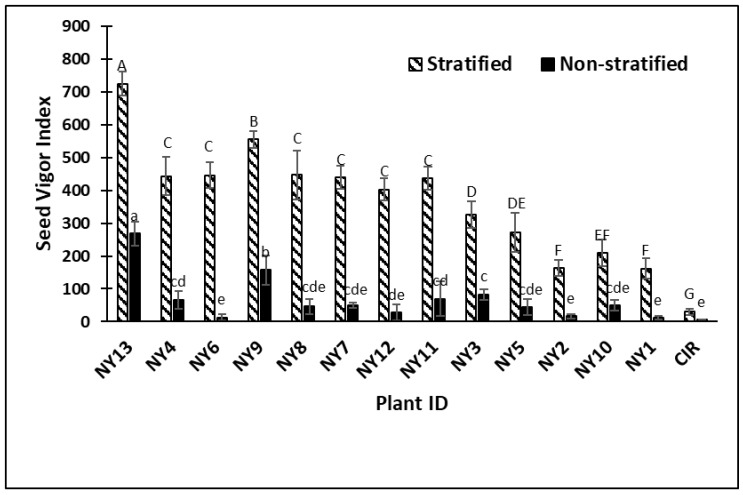
Experiment 2, fall 2017, seed vigor index results showing switchgrass genotypes and the CIR check in stratified and non-stratified germination assays. Capitalized letters represent significant differences among Plant ID genotypes after seed stratification, while lower case letters represent significant differences for non-stratified Plant ID genotypes. Plant IDs with the same letter are not significantly different. Errors bars are standard deviation of means of four replications per Plant ID. *P* < 0.0001 value for both stratified and non-stratified germination with LSD values of 64.4 and 49.4 respectively.

**Figure 3 plants-08-00394-f003:**
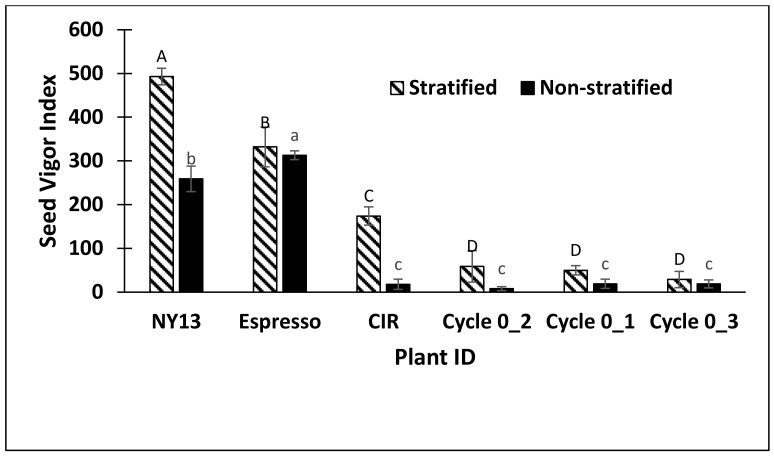
Experiment 3, spring 2018 results showing seed vigor index of switchgrass genotype NY13, Espresso, the CIR check and Cycle 0 in stratified and non-stratified germination assays. Capitalized letters represent significant differences among Plant ID genotypes after seed stratification, while lower case letters represent significant differences for non-stratified Plant ID genotypes. Plant IDs with the same letter are not significantly different. Errors bars are standard deviation of means of four replications per Plant ID. *P* < 0.0001 value for both stratified and non-stratified germination with LSD values of 44.4 and 23.0 respectively.

**Table 1 plants-08-00394-t001:** ID codes and pedigree information for switchgrass genotypes and check cultivars used in the experiments. The Experiments column shows which genotypes and check cultivars were used in each experiment. Plant vigor score where “1” is the least vigorous plant and “5” is the most vigorous plant) and plant height (cm) recorded in the fall of 2015. Parent Location, and Parent Germplasm Resources Information Network (GRIN) Plant Name columns show the location, and GRIN plant name for the parents of the switchgrass genotypes and check cultivars.

ID	Experiments	Plant Vigor	Plant Height	Parent Location	Parent GRIN Plant Name
NY1	1, 2	5	114	Orleans Co, NY	9106194
NY2	1, 2	5	127	Orleans Co, NY	9106194
NY3	1, 2	5	107	Niagara Co, NY	9106191
NY4	1, 2	3	109	Genesee Co, NY	9106195
NY5	1, 2	4	117	Genesee Co, NY	9106195
NY6	1, 2	5	129	Genesee Co, NY	9106195
NY7	1, 2	3	124	Steuben Co, NY	9106189
NY8	1, 2	4	96	Genesee Co, NY	9106195
NY9	2	4	107	Steuben Co, NY	9106189
NY10	2	3	117	Hudson Co, NY	9086098
NY11	2	4	129	Steuben Co, NY	9106189
NY12	2	3	129	Genesee Co, NY	9106195
NY13	1, 2, 3	4	112	Genesee Co, NY	9106195
CIR	1, 2, 3	-	-	Ernst Seed, PA *	Cave-in-Rock
Espresso	3	-	-	Starkville, MS **	
Cycle 0	3	-	-	Multiple locations	

* Cave-in-Rock seed, harvested in 2007, was purchased from Ernst Conservation Seed, Meadsville, Pennsylvania (PA) and was planted in the 2008 and 2009 breeding nurseries. ** Espresso, harvested in 2017, was provided by Dr. Brian Baldwin of Mississippi State (MS) University.

**Table 2 plants-08-00394-t002:** Experiment 1 spring 2016 results, maximum percentage germination (MPG%), days to reach 50% maximum germination (T50) of switchgrass genotypes (ID) and switchgrass cultivar Cave-in-Rock (CIR), from stratified and non-stratified controlled environment assays.

ID	MPG %		T50	
	Stratified	Non-Stratified	Stratified	Non-Stratified
NY7	55 A	13 BC	12 CD	19 ABC
NY3	44 B	18 AB	14 AB	13 BC
NY6	40 BC	9 C	10 D	11 C
NY4	40 BC	20 A	12 CD	24 A
NY8	39 BC	11 C	11 CD	19 ABC
NY1	33 CD	13 BC	16 A	23 AB
NY10	28 D	8 CD	12 BC	17 ABC
NY5	26 D	13 BC	10 CD	17 ABC
NY2	15 E	3 DE	15 A	10 C
CIR	3 F	1 E	12 CD	11 C
*P* value <	0.0001	0.0001	0.0001	0.0001
LSD	10.0	6.1	2.2	9.1

**Table 3 plants-08-00394-t003:** Experiment 2 results, maximum percentage seed germination (MPG), days to reach 50% maximum germination (T50) of switchgrass genotypes (ID) and switchgrass cultivar Cave-in-Rock (CIR), from stratified and non-stratified controlled environment assays.

ID	MPG %		T50	
	Stratified	Non-Stratified	Stratified	Non-Stratified
NY13	67 A	28 A	7 E	9 E
NY4	59 AB	11 CDE	11 BCDE	13 CDE
NY6	58 AB	3 EF	11 BCDE	21 BC
NY9	57 AB	21 AB	8 GH	11 DE
NY8	57 AB	8 DEF	11 BCDE	20 BCD
NY7	55 B	9 DEF	11 BCDE	13 CDE
NY12	52 BC	5 EF	11 BCDE	18 BCD
NY11	50 BC	14 BCD	10 EFG	20 BCD
NY3	45 C	18 BC	11 BCDE	19 BCD
NY5	32 D	9 DEF	9 EFD	19 BCD
NY2	32 D	5 EF	17 A	33 A
NY10	32 D	11 CDE	13 BC	25 AB
NY1	27 D	3 EF	13 BC	26 AB
CIR	4 E	2 F	14 B	32 A
*P* value <	0.0001	0.0001	0.0001	0.0001
LSD	10.4	8.6	2.6	9.1

**Table 4 plants-08-00394-t004:** Experiment 3 results, maximum percentage seed germination (MPG), days to reach 50% maximum germination (T50) of seed harvested in the first seed production year (cycle 0) a selected cold-tolerant genotype (ID) and switchgrass cultivars Cave-in-Rock (CIR) and Espresso from stratified and non-stratified controlled environment assays.

ID	MPG %		T50	
	Stratified	Non-Stratified	Stratified	Non-Stratified
NY13	89 A	78 A	6 C	11 CD
Espresso	68 B	83 A	8 BC	11 D
CIR	51 C	5 B	12 AB	17 AB
Cycle 0_2	16 D	4 B	11 AB	20 A
Cycle 0_1	13 D	8 B	12 AB	17 AB
Cycle 0_3	8 D	7 B	15 A	14 BC
*P* value ≤	0.0001	0.0001	0.004	0.0002
LSD	11.7	6.7	3.9	3.5

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
