# Peer review of "Evaluation of Switchgrass Genotypes for Cold-Tolerant Seed Germination from Native Populations in the Northeast USA"

_plants, 2019, doi:10.3390/plants8100394_

Round 1

Reviewer 1 Report

In General this Manuscript is very interesting and well written. English and style are perfectly fine only couple of minor spell check was found

Line 35 please change indiangrass to Indiangrass

Line 126 Please 201800200 check this number or year 

Line 140 please change 25C to 25C0 (the celesiam sign)

Line 194 please change to due to .....due to

Line 217 seed Viability the Cultivar CIR.... to Seed viability for the cultivar CIR

Author Response

Response to Reviewer 1 Comments

Authors of this manuscript appreciate the positive feedback from Reviewer #1.

Comments and Suggestions for Authors:

In general, this Manuscript is very interesting and well written. English and style are perfectly fine only couple of minor spell check was found

Point 1:Line 35 please change indiangrass to Indiangrass –

Response 1:Corrected Line 42

Point 2:Line 126 Please 201800200 check this number or year –

Response 2:Plant Variety Protection Number to clarify.

Point 3:Line 140 please change 25C to 25C0 (the celesiam sign) 

Response 3:This has been corrected.

Point 4:Line 194 please change to due to .....due to

Response 4:This has been corrected.

Point 5:Line 217 seed Viability the Cultivar CIR.... to Seed viability for the cultivar CIR

Response 5:This has been corrected.

Reviewer 2 Report

Dear Authors,

the subject of this study is of interest both for the scientific field and for the breeders. Unfortunately this version is not acceptable for publication.
Here are some of my suggestions:
TITLE: is it not clear here in the text the term "cold tolerant seed germination"? Please change the title and also the text.
ABSTRACT: both the introduction to the topic, the materials and methods and the purpose of this study are missing. This section is really unclear.
KEYWORDS: I suggest not using the same words used in the title.
INTRODUCTION: Line 34-37 unnecessary. Line 53: the role of hormones and in particular ABA is also very important. Both the hypotheses and the aims of this study are missing.
M&M: this section is not clear to me. It is not clear what the three different experiments are. Perhaps a summary table could be useful. Moreover, in the lines 80-91 it is not clear which seeds are actually used. Line 102: cold ... at what temperature?
The description of the test is not well specified. Why was a moderately sensitive cultivar used? It would have been useful to have also sensitive and tolerant ones. What is meant by "control" and "non-selected control"? The captions of the tables and figures seem more of materials and methods than clear and legible captions. What numbers and treatments have been made? Why was the tetraziolium test done on only a few? vitality should be evaluated first of all.
Line 143-148 are not data analysis. Line 148: what are experiments 1,2 and 3? Have the percentage data been transformed? Was the data normal? Homogeneous?
R&D: the presentation of the data is scarce and not statistically correct. It is not right to write "best cold tolerant ...". There is no discussion !! Statistical analyzes should also be conducted between stratified and non stratified treatment pairs. In the histograms there are no statistics and the ANOVA letters are not explained in the caption.
CONCLUSION: not relevant with the rest of the text. The article provides for germination tests. In the end, what are the best conditions? For your purposes that are not clear to the reader.

Author Response

Response to Reviewer 2 Comments

Authors of this manuscript would like to thank the reviewer 2 for the thoughtful comments and efforts towards improving our manuscript.

Comments and Suggestions for Authors

Dear Authors,

the subject of this study is of interest both for the scientific field and for the breeders. Unfortunately, this version is not acceptable for publication.
Here are some of my suggestions:

Point 1:TITLE: is it not clear here in the text the term "cold tolerant seed germination"? Please change the title and also the text.

Response 1:Title was changed slightly. New title: “Evaluation of Switchgrass Genotypes for Cold Tolerant Seed Germination from Native Populations in the Northeast USA”

Point 2: ABSTRACT: both the introduction to the topic, the materials and methods and the purpose of this study are missing. This section is really unclear.

Response 2:Additional information has been added to the abstract to add clarity for the purpose of the project.

Point 3:KEYWORDS: I suggest not using the same words used in the title.

Response 3:A few keywords were changed.

Keywords: switchgrass; C4grasses; cold tolerance; bioenergy; seed germination; plant breeding

Point 4:INTRODUCTION: Line 34-37 unnecessary.

Response 4:We chose to keep that information for background for individuals not familiar with warm season grasses in the USA.

Point 5: Line 53: the role of hormones and in particular ABA is also very important. Both the hypotheses and the aims of this study are missing.

Response 5:Additional information has been added to the introduction and materials and methods sections to add breadth and clarity.

Point 6:M&M: this section is not clear to me. It is not clear what the three different experiments are. Perhaps a summary table could be useful.

Response 6:Experiments were more clearly defined. Table 1 is the summary table.

Point 7:Moreover, in the lines 80-91 it is not clear which seeds are actually used.

Response 7:Plant lines were renamed to clearly identify which lines were used in each experiment.

Point 8:Line 102: cold ... at what temperature?

Response 8:Temperature has been added.

Point 9:The description of the test is not well specified.

Why was a moderately sensitive cultivar used? It would have been useful to have also sensitive and tolerant ones.

Response 9: A sentence was added to indicate enough seed was available for the CIR control.

Point 10:What is meant by "control" and "non-selected control"?

Response 10:Non-selected control indicates the line was not evaluated in preliminary screening. Line 339-340.

Point 11:The captions of the tables and figures seem more of materials and methods than clear and legible captions.

Response 11:Captions were amended to add clarity.

Point 12:What numbers and treatments have been made? Why was the tetraziolium test done on only a few? vitality should be evaluated first of all.

Response 12:TZ test was limited by the numbers of seeds available for analysis.

Additional information has been added to the introduction and materials and methods sections to add breadth and clarity.

Point 13:Line 143-148 are not data analysis.

Response 13:This section was moved.

Point 14:Line 148: what are experiments 1,2 and 3? Have the percentage data been transformed? Was the data normal? Homogeneous?

Response 14:Additional information on data analysis and normality of data has been provided.

Point 15:R&D: the presentation of the data is scarce and not statistically correct. It is not right to write "best cold tolerant ...". There is no discussion !!  

Response 15:Results and Discussion section is combined there is a separate Conclusions section.

Point 16:Statistical analyzes should also be conducted between stratified and non stratified treatment pairs.

Response 16:Comparisons were analyzed by plant genotypes not stratification.

Point 17:In the histograms there are no statistics and the ANOVA letters are not explained in the caption.

Response 17:Statistics are provided in the caption, however for clarity ANOVA letters were added to the figures.

Point 18: CONCLUSION: not relevant with the rest of the text. The article provides for germination tests. In the end, what are the best conditions? For your purposes that are not clear to the reader.

Response 18: Additional information has been added to the Conclusion section to provide background on the purpose and potential use of cold tolerant lines for production and genetic screening.

Reviewer 3 Report

Switchgrass genotypes for cold-tolerant seed germination can be used as a resource for the breeding programs to create new and better varieties. These varieties may be used in feedstock for biomass energy production, soil conservation, forage, etc.
However, I am unable to understand the purpose of the study as it is not stated anywhere in the manuscript, please elaborate on it.

What kind of questions can be answered using this information, please shed some light on them as well?

Why does the author not provide some information about the genetic variability among these genotypes?

Please, consider a better word for "experiment 1 and 2" because it is a little confusing to me, and seems as I am reading a report, not a peer-reviewed manuscript.

Lines 209 to 211 have a formatting issue, please, fix it.

Author Response

Responses to Reviewer 3 Comments

We would like to thank the reviewer 3 for the valuable comments and suggestions.

Comments and Suggestions for Authors

Switchgrass genotypes for cold-tolerant seed germination can be used as a resource for the breeding programs to create new and better varieties. These varieties may be used in feedstock for biomass energy production, soil conservation, forage, etc.

Point 1:However, I am unable to understand the purpose of the study as it is not stated anywhere in the manuscript, please elaborate on it.

Response 1:Lines 75-77 were moved to the abstract to add clarity to this project. Additional information on selection for cold tolerance has also been added to the introduction lines 83-89 and Conclusion sections 251-278.

Point 2:What kind of questions can be answered using this information, please shed some light on them as well?

Response 2: One focus of the project was to determine if selection of cold tolerant genotypes was possible. Another is investigation of genetic mechanisms involved with cold tolerant germination. This has been added to the text.

Point 3:Why does the author not provide some information about the genetic variability among these genotypes?

Response 3: Our project did not focus on genetic variation.

Point 4: Please, consider a better word for "experiment 1 and 2" because it is a little confusing to me, and seems as I am reading a report, not a peer-reviewed manuscript.

Response 4:Slight changes were made to names of the experiments “spring 17” fall 2017”.

Point 5:Lines 209 to 211 have a formatting issue, please, fix it.

Response 5:The formatting was corrected.

Reviewer 4 Report

This manuscript describes a study on investigation the germination of cold tolerant and commercial switchgrass. It may be potentially useful for researchers in studying switchgrass germination, but it is also limited to extract too much effective information. I have several suggestions for improving this manuscript as below.

The abbreviation of Northeast as NE is confusing, since it is easily mixed with Nebraska. The description in the methods “2.1 Switchgrass genotypes” is not very concise, please provide an additional supplemental figure to illustrate your genotype selection. Table 2, 3 and 4 are not very straightforward, why not use a figure to represent the result and clearly write about which test you used and which two pairs are compared? It seems there is no figures about seed vigor mentioned in any of three tables, but it was mentioned in the figure legend. Please indicate how many replicates you used in the figure legend of figure 1; on y-axis of figure 1, please label axis ticks. The same for figure 2 and 3. I think the current name for switchgrass genotype is too long and not informative, it makes more easier to identify genotypes if you just use something like “SW1” to represent a genotype like 11-34-8-2014, and give a supplemental table to refer each one. Although I understand the current name maybe easier for authors to identify, but it is really easy to messy up when readers read it. For the seed from the same parent, it will also be meaningful to compare and see how different or similar they are.

Author Response

Responses to Reviewer 4 Comments

We thank the Reviewer 4 for the careful and insightful review of our manuscript.

Comments and Suggestions for Authors

This manuscript describes a study on investigation the germination of cold tolerant and commercial switchgrass. It may be potentially useful for researchers in studying switchgrass germination, but it is also limited to extract too much effective information. I have several suggestions for improving this manuscript as below.

Point 1:The abbreviation of Northeast as NE is confusing, since it is easily mixed with Nebraska.

Response 1:Northeast is spelled out in the manuscript to avoid confusion.

Point 2:The description in the methods “2.1 Switchgrass genotypes” is not very concise, please provide an additional supplemental figure to illustrate your genotype selection.

Response 2:Two additional figures were added and table was added to add clarity to how genotypes were selected. Additional information was also added to the materials and methods section. Lines 283-330 were rewritten.

Point 3:Table 2, 3 and 4 are not very straightforward, why not use a figure to represent the result and clearly write about which test you used and which two pairs are compared?

Response 3:Tables were reformatted for clarity.

Point 4: It seems there is no figures about seed vigor mentioned in any of three tables, but it was mentioned in the figure legend.

Response 4:The mention of seed vigor index has been deleted from the table captions.

Point 5:Please indicate how many replicates you used in the figure legend of figure 1; on y-axis of figure 1, please label axis ticks. The same for figure 2 and 3.

Response 5:The figures have been corrected.

Point 6: I think the current name for switchgrass genotype is too long and not informative, it makes easier to identify genotypes if you just use something like “SW1” to represent a genotype like 11-34-8-2014, and give a supplemental table to refer each one. Although I understand the current name maybe easier for authors to identify, but it is really easy to messy up when readers read it. For the seed from the same parent, it will also be meaningful to compare and see how different or similar they are. 

Response 6:The genotypes have been renamed NY1-NY13.

Round 2

Reviewer 2 Report

Dear Authors,

although the present form is improved in many of its parts, I believe that this work is still confused and it seems that many data from different tests have been inserted. To increase the quality of work and make it acceptable, I suggest verifying which data are really useful to you and making the text more readable. Probably some accessions and some timing can be eliminated. For example, the abstract remains unclear. The theses that have been studied are not specified, the m & ms are missing and in the end the results are not understood.

Author Response

Response to Reviewer 2 Comments

The author thanks Reviewer #2 for the valuable comments and positive feedback on our first and the current revised draft of this manuscript.

Point 1: Dear Authors, although the present form is improved in many of its parts, I believe that this work is still confused and it seems that many data from different tests have been inserted.

 Response 1: The other reviewers asked for additional background information on the switchgrass nurseries to provide information on how selections were made for the cold tolerant germination lines discussed in this manuscript. This material was added as supplemental figures and an additional table was added. The materials and methods and conclusion sections have been edited to add clarity.

 Point 2: To increase the quality of work and make it acceptable, I suggest verifying which data are really useful to you and making the text more readable. Probably some accessions and some timing can be eliminated.

Response 2:Information added as the supplemental material was to provide background on switchgrass selections. The focus of this research manuscript is the three experiments reported and discussed in the Tables 1-4 and figures 1-3. Some text in the materials and methods section was removed.

Point 3: For example, the abstract remains unclear. The theses that have been studied are not specified, the m & ms are missing and in the end the results are not understood.

 Response 3: The abstract and conclusions sections have been amended for clarity. The M&M section was moved to follow the results & discussion section as specified by the journal instructions. Some changes were also made to the M&M sections for clarity.

Reviewer 4 Report

Authors addressed all of my concerns properly and I think the current manuscript is ready for publishing. 

Author Response

Point 1: “Authors addressed all of my concerns properly and I think the current manuscript is ready for publishing.”

Response 1:Authors of this manuscript appreciate the positive feedback from Reviewer #4.
